# Multiple Primary Malignancies in Patients with Gynecologic Cancer

**DOI:** 10.3390/jcm11010115

**Published:** 2021-12-26

**Authors:** Eun-Jung Yang, Ji-Hyeon Lee, A-Jin Lee, Nae-Ry Kim, Yong-Taek Ouh, Mi-Kyung Kim, Seung-Hyuk Shim, Sun-Joo Lee, Tae-Jin Kim, Kyeong-A So

**Affiliations:** 1Department of Obstetrics and Gynecology, KonKuk University Hospital, Seoul 05030, Korea; manim486@hanmail.net (E.-J.Y.); 20190168@kuh.ac.kr (J.-H.L.); 20170050@kuh.ac.kr (A.-J.L.); 20190125@kuh.ac.kr (N.-R.K.); 20130131@kuh.ac.kr (S.-H.S.); lsj1121@kuh.ac.kr (S.-J.L.); 20190002@kuh.ac.kr (T.-J.K.); 2Department of Obstetrics and Gynecology, Grauate School of Medicine, Kangwon National University, Chuncheon 24289, Korea; oytjjang@gmail.com; 3Department of Obstetrics and Gynecology, Ewha Womans University Mokdong Hospital, Seoul 07804, Korea; asterik79@gmail.com

**Keywords:** multiple primary malignant tumors, gynecologic cancers, synchronous malignancies, metachronous malignancies

## Abstract

Objective: To investigate the prevalence and oncologic outcomes of patients with multiple primary malignant tumors (MPMT) with gynecologic cancer. Methods: This retrospective study included 1929 patients diagnosed with gynecologic cancer at a tertiary medical center between August 2005 and April 2021. The clinical data included cancer location, age at primary malignancy diagnosis, interval between primary and secondary cancer, stage of cancer, family history of cancer, genetic testing, dates of last follow-up, recurrence, and death. Results: The prevalence of MPMT with gynecologic cancer in patients was 8.6% and the mean diagnostic period between primary and secondary cancer was 60 months. Furthermore, 20 of the 165 patients with MPMT had multiple primary gynecologic cancers (MPGC), whereas 145 had gynecologic cancer coexisting with non-gynecologic cancer (GNC). Endometrial-ovarian cancer (60%) was the most common coexisting cancer in the MPGC group, whereas the most common non-gynecologic cancer in the GNC group was breast cancer (34.5%). There were 48 patients with synchronous cancer and 117 patients with metachronous cancer. The incidence of synchronous cancer was higher in the MPGC group than in the GNC group (*p* = 0.037). Significantly more patients had early-stage ovarian cancer in the MPGC group than in the GNC group (*p* = 0.031). The overall recurrence and mortality rates were 15.8% and 8.5%, respectively, in patients with MPMT. Conclusion: Synchronous cancer incidence was significantly higher in the MPGC than in the GNC group. Early-stage ovarian cancer was more highly diagnosed in patients with MPGC than in those with GNC. A systematic examination after primary cancer diagnosis could facilitate the early diagnosis of secondary primary malignancy, thereby improving patient prognosis.

## 1. Introduction

Gynecologic cancers consist of cancers that originate in the female reproductive system, the most common of which is uterine cancer, and these cancers often have multiple primary malignant tumors (MPMT). MPMT was first described in 1932 by Warren and Gates [1] who proposed the following definition as one or more tumors (1) all diagnosed as cancer, (2) all with different pathological origins, (3) and due to metastasis or recurrence are excluded [1]. Previous studies reported various MPMT prevalence rates as 0.4–21.0% [2,3,4,5], and the proportion of MPMT associated with gynecologic cancers was reported to be 1.9–4.3% [6,7].

MPMT with gynecologic cancer can be divided into those with multiple primary gynecologic cancer (MPGC) and gynecologic cancer with non-gynecologic cancer (GNC) according to the type of comorbid cancer. GNC in patients is associated with breast [8,9,10,11], colorectal [9,10,11,12,13], bladder [9,11,13], kidney [11], and lung [12] cancers. In patients with MPGC, a combination of endometrial-ovarian cancer was the most common [14,15]. MPMT is further divided into synchronous or metachronous malignancy according to the time interval between the dates of diagnosis of one or more primary tumors [2]. A synchronous malignancy is defined as one where the diagnosis of the subsequent malignant tumor is made concurrently or within 6 months from the diagnosis date of the first primary malignancy. A metachronous malignancy refers to a diagnosis of a subsequent malignant tumor made >6 months after the first primary malignancy diagnosis [16,17,18]. Furthermore, 34–38% and 62–66% of MPMT have been reported to be synchronous and metachronous malignancies, respectively [2].

Treatment of MPMT requires the cautious and meticulous selection of appropriate approaches and treatment strategies. Treatment of the first primary malignancy should be planned to ensure that subsequent malignancies are not adversely affected by increased toxicity or pharmacological interactions [19]. Early diagnosis of subsequent malignancies is important because delayed diagnosis worsens the patient’s prognosis. Currently, few studies have focused on MPMT in gynecologic cancer and, therefore, this study was conducted to investigate the clinical features and oncologic outcomes of MPMT in patients with gynecologic cancer.

## 2. Materials and Methods

In this study, which was approved by our institutional review board (IRB No. KUMC 2021-06-041), we retrospectively reviewed the medical records of 1929 patients with gynecologic cancers who presented at tertiary institution medical center between August 2005 and April 2021. All patient information and clinical data were archived in a central database at our institution. All patients diagnosed with MPMT with gynecologic cancer since the opening of our institute in August 2005 were enrolled in the study, except for those excluded based on the diagnostic criteria of Warren and Gates [1].

Gynecologic cancers considered in this study were cervical, ovarian, endometrial, and vulvovaginal cancers. Patients who met the study inclusion criteria were divided into the following two groups, according to the origin of the accompanying malignancy and the diagnosis period, which were comparatively analyzed. (1) Those diagnosed with one or more gynecologic cancers of different origins (MPGC group) or with gynecologic cancer coexisting with non-gynecologic cancer (GNC group) and (2) synchronous or metachronous cases based on a 6 month period between diagnosis dates.

The clinical data analyzed were collected by reviewing medical charts and consisted of cancer location, age at primary malignancy diagnosis, interval between primary and secondary cancer, stage of cancer, family history of cancer, genetic testing, and dates of last follow-up, recurrence, and death. Clinical analysis of the data in the study was conducted by classifying MPGC and GNC according to the accompanying primary cancer.

In addition, the tumors were further classified into synchronous and metachronous malignancies according to the diagnosis interval of primary cancer. The diagnosis date was determined when the histopathological diagnosis was confirmed. Progression-free survival (PFS) was defined as the duration from the date of diagnosis to the date of relapse or censoring, whereas overall survival (OS) was defined as the duration from the date of diagnosis to the date of death, last follow-up, or censoring.

All analyzes were performed using the statistical package for the social sciences (SPSS) software program version 22.0 (IBM SPSS Statistics, Chicago, IL, USA). Frequency distributions were analyzed using the chi-squared and Fisher’s exact tests. Qualitative data are shown as frequencies and percentages, whereas quantitative data are presented as means ± standard deviation (SD) or median and range. The Kaplan–Meier survival analysis was used to determine the PFS and OS. The results were compared using the log-rank test and a *p*-value ≤ 0.05 was considered statistically significant.

## 3. Results

### 3.1. Site Distribution of Multiple Primary Malignant Tumors

During the study period, 1929 patients with gynecologic cancer were enrolled, including 165 (8.6%) who were diagnosed to have MPMT. In addition, 153 (92.7%) and 12 (7.3%) of the patients with MPMT had double and triple primary malignancies, respectively. The median follow-up period was 101.5 (range, 2–473) months. Furthermore, the mean age at diagnosis of the first and second primary cancer was 51.6 and 56.0 years, respectively. The most common site of the first primary cancer was the cervix (21%), followed by the breast (19%), ovaries (17%), uterus (16%), thyroid (11%), colon/rectum (11%), and others (5%), identified in that order (Figure 1A). Other sites of the first primary cancer were the tonsils, skin, thighs, oral cavity, pancreas, kidneys, lungs, liver, bladder, hematologic, vagina, and stomach. The most common second primary cancer sites are listed in descending order of magnitude: uterus, ovary, cervix, thyroid, breast, colon/rectum, and others. (Figure 1B). The frequencies of the subsequent cancers according to primary cancer type are shown in Figure 2. Thyroid cancer was the most common second primary cancer in cervical cancer. Breast cancer occurred with the highest frequency as a subsequent cancer to ovarian cancer, whereas with endometrial cancer, breast and colorectal cancers had a high frequency as second primary cancers. Endometrial cancer was the most common second primary cancer following breast cancer.

### 3.2. Gynecologic Cancer Coexisting with Non-Gynecologic Cancer (GNC) and Multiple Primary Gynecologic Cancers (MPGC)

Of the 165 patients, 145 (87.9%) and 20 (12.1%) were diagnosed with GNC and MPGC, respectively. Furthermore, in the MPGC group, ovarian-endometrial cancer was the most common (12 patients), followed by ovarian-cervical cancer (4 patients), cervical-endometrial cancer (2 patients), and cervical-vaginal cancer (2 patients). The most common non-gynecologic cancer associated with gynecologic cancer in the GNC group was breast cancer (50/145, 34.5%). There were 33 (22.7%) patients diagnosed with breast cancer as the first cancer before gynecologic cancer, while there were 17 (11.7%) patients diagnosed with breast cancer as the subsequent cancer to gynecologic cancer in the GNC group. There were 13 patients with breast-endometrial cancer, 10 patients with breast-ovarian cancer, and 10 patients with breast-cervical cancer. Furthermore, 7 of the 13 patients with breast-endometrial cancer received anti-estrogen therapy after breast cancer surgery.

The comparison of the clinical characteristics between the GNC and MPGC groups shown in Table 1. There were no significant differences in the age at the time of first cancer diagnosis and the diagnostic period between first and second cancers between both groups. However, 50% of patients in the MPGC group were diagnosed with synchronous malignancies, which was a significantly higher percentage than the 26.2% of patients in the GNC group (*p* = 0.037). The synchronous malignancies in the MPGC group consisted of 8 and 2 cases of ovarian-endometrial and endometrial-cervical cancers, respectively. When analyzed according to the primary gynecology cancer site, ovarian cancer showed a higher rate of early-stage disease in patients with MPGC than in those with GNC (87.5% vs. 52.4%, *p* = 0.031). However, there was no significant difference in early-stage disease between the GNC group and the MPGC group in endometrial cancer and cervical cancer. The overall recurrence rate was 15.8% (26/165) in patients with MPMT. During the study period, 21 (14.5%) and 5 (25.0%) patients in the GNC and MPGC groups relapsed, respectively. According to the primary cancers, the recurrence rate were 76.9% and 23.1% in gynecologic cancer and non-gynecologic cancer, respectively. The mortality rate was 8.5% (14/165) with 13 patients from the GNC group and 1 patient from the MPGC group. The distribution of the direct cause of death was gynecologic (64.3%) and non-gynecologic cancer (35.7%), with ovarian cancer as the most common cause of death. The 5-year PFS and 5-year OS were not significantly different between the GNC and MPGC groups (*p* = 0.166 and *p* = 0.865, respectively, Figure 3A,B).

### 3.3. Synchronous and Metachronous Malignancies

Forty-eight and 117 patients had synchronous and metachronous malignancies, respectively (Table 2). The mean diagnostic interval between the first and second primary cancers in the metachronous malignancies group was 60 months. The most synchronous malignancies were ovarian-endometrial cancer in 8 patients, followed by ovarian-breast cancer in 7 patients, endometrial-breast cancer in 5 patients, and ovarian-thyroid cancer in 5 patients. The mean age at diagnosis for the first primary cancer was 55.4 years and 50.2 years in the synchronous and metachronous malignancies groups, respectively. The metachronous malignancies group showed a tendency to be diagnosed with cancer at a younger age than the synchronous malignancies group (*p* = 0.050). The age at diagnosis was not different according to the types of gynecologic cancer. However, more patients with cervical cancer were diagnosed at the early stage in the metachronous malignancies group than in the synchronous malignancies group (72.3% vs. 63.6%, *p* = 0.015). There were no significant differences between the stage at diagnosis for ovarian and endometrial cancers between the two groups. There was no difference in the 5-year PFS and OS between the synchronous and metachronous malignancies groups (*p* = 0.189 and *p* = 0.152, respectively, Figure 3C,D).

### 3.4. Multiple Primary Malignancies in Gynecologic Patients with Genetic Testing 

Genetic testing has been conducted at our medical institution for hereditary nonpolyposis colorectal cancer (HNPCC) since 2007 and the breast cancer gene (BRCA) since 2017. Nineteen patients underwent the genetic test during the study period and 10 (52.6%) showed positive results as follows: 7 HNPCC, 2 BRCA1, and 1 BRCA2 mutation, respectively. All 10 patients with mutations confirmed by genetic testing were in the GNC group and the results are shown in Table 3.

## 4. Discussion

MPMT with gynecologic cancers was identified at a prevalence rate of 8.6% in this study, which is higher than that in previous studies (1.9–4.3%) [6,7]. This observation suggests that the incidence of MPMT is increasing in relation to several factors, including increased life expectancy, effective screening, accurate diagnosis of malignancies, extensive systematic evaluation for first primary cancers, and long-term follow-up after cancer treatment [20]. Among patients with MPMT with gynecologic cancers, the number of those with GNC was approximately seven times greater than those with MPGC (87.9% vs. 12.1%, respectively) in this study. There were no significant differences in age at first cancer diagnosis, diagnostic interval between primary cancers, recurrence rate, and mortality rate between the GNC and MPGC groups. However, the MPGC group had a higher rate of synchronous malignancies than that of the GNC group (50.0% vs. 26.2%, respectively, *p* = 0.037). In addition, ovarian cancer showed a tendency to be diagnosed at an early stage when it was accompanied by gynecologic cancer but not when accompanied by non-gynecologic cancer (87.5% vs. 52.4%, *p* = 0.031). Several reasons could explain the differences between the GNC and MPGC groups. First, second primary gynecologic cancers may have been fortuitously diagnosed during the surgical staging operation of primary gynecologic cancer. The surgical staging operation usually involves a hysterectomy and salpingo-oophorectomy, which can lead to the detection of a second symptomless primary gynecologic cancer at an early stage. Second, these results could also be explained by the incidental discovery of an asymptomatic second primary gynecologic cancer through hemodynamic testing, computed tomography (CT), and magnetic resonance imaging (MRI). These tests are routinely performed for preoperative tumor evaluation when a patient is diagnosed with primary cancer in our institute. The results of the data analysis of the patients in the MPGC group at our medical center confirmed that 80% were diagnosed with the second primary gynecologic cancer after surgery for primary gynecologic cancer. Furthermore, 20% of patients were diagnosed with preoperative tumor evaluation before primary gynecologic cancer surgery.

Breast cancer was the most common non-gynecologic cancer in the GNC group. Previous studies showed that breast cancer had a higher rate of occurrence as the first primary cancer before the diagnosis of gynecologic cancer than it did as a second diagnosed primary cancer following gynecologic cancer [21,22]. In this study, as a second primary gynecologic cancer associated with breast cancer, endometrial cancer occurred in 13 patients, including 7 (53.8%) with a history of anti-estrogen therapy. Tamoxifen has an anti-estrogenic effect on breast tissue, but has an estrogenic effect on the uterus, and increases the risk of endometrial cancer by 2 to 7 times with long-term use [23]. In addition, high-grade or high-risk types of endometrial cancer are more developed in breast cancer patients after tamoxifen treatment [21]. Therefore, surveillance for other cancers is necessary for patients with breast cancer during anti-estrogen therapy.

Previous studies reported that the proportion of patients with MPGC was higher than that of patients with GNC, and the rate of diagnosis at the early stage of gynecologic cancer was high [20]. There was no difference in the 5-year OS, but the 5-year PFS was confirmed to be longer in the GNC group than it was in the MPGC group [20]. However, there was no difference between the 5-year PFS and OS in this study. The survival outcome results of the patients in this study were different from the results of previous studies according to the distribution of patients with MPMT with gynecologic cancer in our institution. Compared with the findings of previous studies, those of our study suggest that early diagnosis of cancer is an important factor in determining the prognosis of cancer in patients with MPMT.

The synchronous group in which the subsequent cancer was diagnosed within 6 months after the primary cancer was predicted to have a higher early diagnosis rate than that of the metachronous group. However, in patients with cervical cancer, the early diagnosis rate was higher in the metachronous group than it was in the synchronous group (*p* = 0.015). This was likely because the early detection of cervical cancer is increasing and the number of patients in the metachronous group was relatively higher than that in the synchronous group. The diagnostic interval between primary cancers of different origins of metachronous malignancy has been found to be 3–10 years [21,24,25]. The second cancer was diagnosed an average of 5 years after the diagnosis of primary gynecologic cancer in this study. The result shows that attention should also be focused on the long-term surveillance of second cancers in patients with gynecology cancer. Gynecologic cancers, in particular, cause difficulty for women of reproductive age to remain fertile [26]. Maintaining fertility in women of reproductive age can contribute greatly to a patient’s quality of life [27,28]. In this study, 68 (41.2%) patients with gynecological cancer were diagnosed with a first primary cancer before menopause. If a woman with reproductive potential is diagnosed with gynecological cancer at an early stage, measures to maintain fertility should be considered in accordance with the guidelines for preserving fertility [29,30].

Previous studies have shown that genetic mutations cause MPMT [31,32,33]. Approximately 5% and 10% of endometrial and ovarian cancer, respectively are caused by genetic predisposition [31,32]. Women with the BRCA1 mutation have a 39–46% and 65–85% risk of developing ovarian and breast cancers, respectively [33]. The reported risk of ovarian and breast cancers in women with a BRCA2 mutation is 10–27% and 45–85%, respectively [33]. Lynch syndrome, also known as HNPCC syndrome, accounts for most hereditary endometrial cancers. Lynch syndrome is most commonly associated with colorectal and endometrial cancer, with a lifetime risk of 40–60% [33]. MPMT in gynecologic cancers may be caused by a genetic predisposition and genetic testing can predict the likelihood of subsequent cancers. In this study, the genetic testing rate was low because of its late introduction to our hospital. Genetic testing should be considered because it facilitates the prediction of the likelihood of MPMT, improves patient prognosis, and has preventive effects.

Although this study has limitations as a retrospective study conducted at a single medical institution, it also has the following several strengths. First, this study was conducted on all patients with MPMT associated with gynecologic cancers for >10 years in real clinical practice. Second, we identified the characteristics of each group by distinguishing the diagnosis time and origin of MPMT in patients with gynecologic cancer. Further large-scale prospective studies with long-term follow-up data are needed to evaluate the clinical implications of multiple primary malignancies. 

In conclusion, systematic examination after diagnosis of primary gynecologic cancer would facilitate the diagnosis of second primary cancer. Because secondary malignancies are diagnosed an average of 5 years after the primary malignancies, attention should be paid to long-term surveillance of secondary malignancies in patients with gynecologic cancer. In addition, because there may be a risk of secondary malignancy following the treatment of primary cancer, it is important to monitor the occurrence of additional cancer in patients receiving cancer treatment. Although no significant difference in survival outcome were identified in this study, early diagnosis of MPMT is thought to improve the prognosis of patients and help to preserve the fertility of women of reproductive age. The results of this study may be useful in the early diagnosis of MPMT in patients with gynecologic cancer and may contribute to developing strategies to conduct surveillance of patients with gynecologic cancer to monitor the risk of a second malignancy.

## Figures and Tables

**Figure 1 jcm-11-00115-f001:**
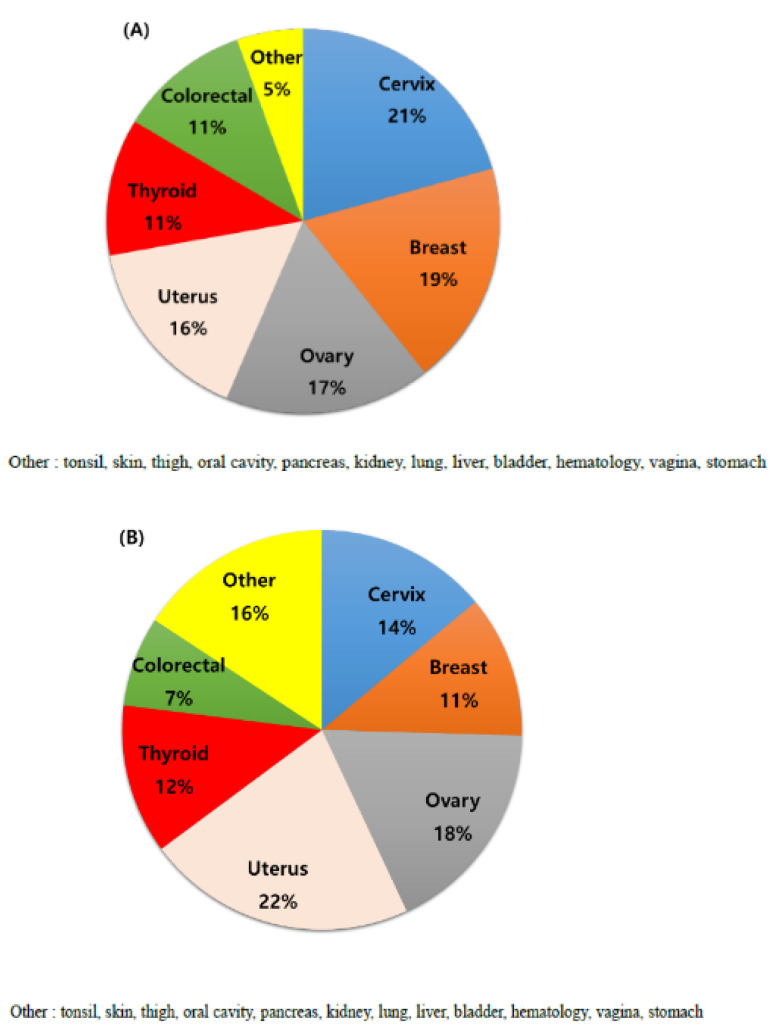
Site distribution of primary cancer. (**A**) First and (**B**) second primary cancers.

**Figure 2 jcm-11-00115-f002:**
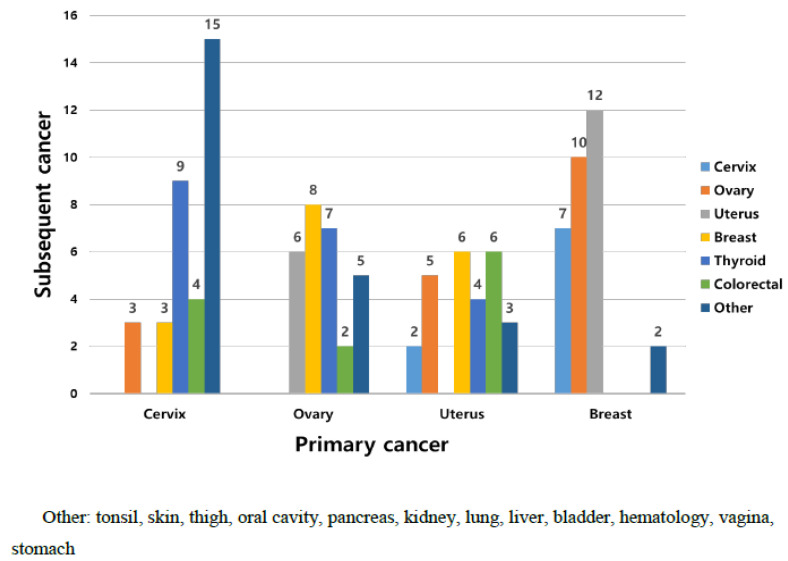
Frequency of subsequent cancer according to primary cancer type.

**Figure 3 jcm-11-00115-f003:**
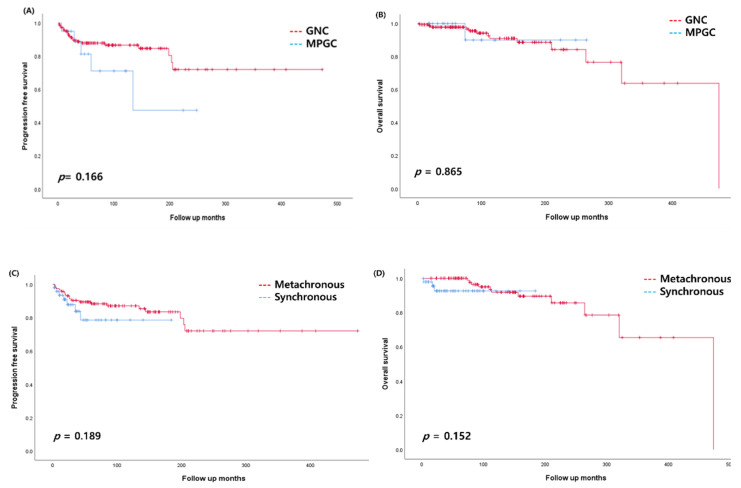
Progression-free survival (PFS) and overall survival (OS) of patients with multiple primary malignant tumors (MPMT) with gynecologic cancer. Comparisons of (**A**) PFS and (**B**) OS between patients with multiple primary gynecologic cancer (MPGC) and gynecologic cancer coexisting with non-gynecologic cancer (GNC). Comparison of (**C**) PFS and (**D**) OS between patients with synchronous and metachronous malignancies.

**Table 1 jcm-11-00115-t001:** Comparison of patient characteristics between non-gynecologic cancer (GNC) and multiple primary gynecologic cancers (MPGC).

Variables	GNC (N = 145)	MPGC (N = 20)	*p*-Value
Age at diagnosis of first cancer, yr	52.3 ± 12.4	45.7 ± 9.3	0.377
Interval between 1st and 2nd cancer, mo	41.5 (0–420)	5.5 (0–241)	0.992
Synchronous cancer, *n* (%)	38 (26.2%)	10 (50.0%)	0.037
Stage of OC, *n* (%)			0.031
I-II	22 (52.4)	14 (87.5)	
III-IV	16 (38.1)	1 (6.3)	
unknown	4 (9.5)	1 (6.3)	
Stage of EC, *n* (%)			0.604
I-II	35 (72.9)	12 (85.7)	
III-IV	12 (25.0)	2 (14.3)	
unknown	1 (2.1)	0	
Stage of CC, *n* (%)			
I-II	39 (72.2)	5 (62.5)	0.449
III-IV	6 (11.1)	0	
unknown	9 (16.7)	3 (37.5)	
Died of disease, *n* (%)	13 (9.0)	1 (5.0)	1.000
Recurrence of disease, *n* (%)	21 (14.5)	5 (25.0)	0.321

Abbreviations: GNC, gynecologic cancer coexisting with non-gynecologic cancer; MPGC, multiple primary gynecologic cancer; OC, ovarian cancer; EC, endometrial cancer; CC, cervical cancer.

**Table 2 jcm-11-00115-t002:** Patient characteristics compared between synchronous malignancies and metachronous malignancies.

Variables	Synchronous Malignancies (N = 48)	Metachronous Malignancies (N = 117)	*p*-Value
Age at diagnosis of first cancer, yr	55.4 ± 12.9	50.2 ± 11.7	0.050
Stage of OC, *n* (%)			0.817
I-II	11 (64.7)	17 (47.2)	
III-IV	5 (29.4)	12 (33.3)	
unknown	1 (5.9)	3 (8.3)	
Stage of EC, *n* (%)			0.392
I-II	12 (63.2)	28 (73.7)	
III-IV	6 (31.6)	10 (26.3)	
unknown	1 (5.3)	0	
Stage of CC, *n* (%)			0.015
I-II	7 (63.6)	34 (72.3)	
III-IV	4 (36.4)	3 (6.4)	
unknown	0	10 (21.3)	

Abbreviations: OC, ovarian cancer; EC, endometrial cancer; CC, cervical cancer.

**Table 3 jcm-11-00115-t003:** Multiple primary malignancies in patients with gynecologic cancer with positive genetic test results (*n* = 10).

Genetic Test Result	FHC	First Caner, Age at Diagnosis	Recurrence PFS, mo	Status, Time to Survival, mo	Second Cancer, Age at Diagnosis, mo	Recurrence, PFS, mo	Status, Time to Survival, mo
HNPCC	No	Endometrium 37	No	Remission 10	Colorectal 37	No	Remission 9
HNPCC	No	Endometrium 64	No	Remission 82	Colorectal 64	No	Remission 81
HNPCC	Yes	Colorectal 35	Yes 45	Remission 52	Endometrium 48	Lost to FU	Lost to FU 0
HNPCC	Yes	Colorectal 31	No	Lost to FU 164	Endometrium 44	Lost to FU	Lost to FU 2
HNPCC	No	Colorectal 60	No	Remission 43	Cervix 61	No	Remission 28
HNPCC	No	Cervix 50	No	Remission 23	Colorectal 50	No	Remission 22
HNPCC	No	Colorectal 41	Yes 60	Remission 325	Ovary 48	No	Lost to FU 241

Abbreviations: PFS; progression-free survival, FHC, Family history of cancer; HNPCC, hereditary nonpolyposis colorectal cancer; BRCA, breast cancer gene; FU, follow-up.

## Data Availability

All the studies used in this study are published in the literature.

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
