# Peer review of "Multiple Primary Malignancies in Patients with Gynecologic Cancer"

_jcm, 2021, doi:10.3390/jcm11010115_

Round 1
Reviewer 1 Report
More genetic testing would be interesting. The results must be better systematized, because those paragraphs are difficult to be read and understood.
Author Response
More genetic testing would be interesting. The results must be better systematized, because those paragraphs are difficult to be read and understood.
→ I deeply appreciate that you made a good point. I absolutely agree with the reviewer’s comments. It is believed that as genetic testing evolves in a variety of ways, more genetic testing will be performed to diagnose MPMT patients. Although the genetic test rate is on the low side due to the delayed introduction of the test at this institute, it is expected that the research will proceed with better data if this part is improved in the future.
→ The results section was revised to make it easier to read and understood. And the manuscript was additional edition of the English language by Editage.
Reviewer 2 Report
I read with great interest the manuscript, which falls within the aim of this Journal. In my honest opinion, the topic is interesting enough to attract the readers’ attention. Nevertheless, authors should clarify some points and improve the discussion, as suggested below.
Authors should consider the following recommendations:
- Manuscript should be further revised in order to correct some typos and improve style.
- In light of the advanced techniques to detect early stage disease, to date it is mandatory to consider even the possibility of fertility-sparing approaches in order to preserve reproductive potential of women affected by gynecological cancers. I invite authors to discuss this point, referring to: PMID: 28868252; PMID: 28840513.
Author Response
I read with great interest the manuscript, which falls within the aim of this Journal. In my honest opinion, the topic is interesting enough to attract the readers’ attention. Nevertheless, authors should clarify some points and improve the discussion, as suggested below.
Authors should consider the following recommendations:
Manuscript should be further revised in order to correct some typos and improve style.
→ I deeply appreciate that you made a good point. The manuscript was additional edition of the English language by Editage.
In light of the advanced techniques to detect early stage disease, to date it is mandatory to consider even the possibility of fertility-sparing approaches in order to preserve reproductive potential of women affected by gynecological cancers. I invite authors to discuss this point, referring to: PMID: 28868252; PMID: 28840513.
→ I completely agree with your opinion. I replenished the discussion section, including the fertility-sparing approaches about what you pointed out (pages 9, lines 256-262, pages 10, lines 290-291).